# Sequence analysis of sickness absence and disability pension days in 2012–2018 among privately employed white-collar workers in Sweden: a prospective cohort study

Laura Salonen,[1] Kristina Alexanderson,[2] Kristin Farrants  [2]

¹Finnish Institute of Occupational Health, Helsinki, Finland
²Karolinska Institutet Division of Insurance Medicine, Stockholm, Sweden

**Correspondence to**
Dr Kristin Farrants;
kristin.farrants@ki.se

## ABSTRACT

**Objective** The aim of the study is to explore sequences of sickness absence (SA) and disability pension (DP) days from 2012 to 2018 among privately employed white-collar workers.

**Design** A 7-year prospective cohort study using microdata from nationwide registers.

**Setting** Sweden.

**Participants** All 1 283 516 privately employed white-collar workers in Sweden in 2012 aged 18–67.

**Methods** Sequence analysis was used to describe clusters of individuals who followed similar development of SA and DP net days/year, and multinomial logistic regression to analyse associations between sociodemographic variables and belonging to each observed cluster of sequences. Odds ratios (ORs) and 95% confidence intervals (CIs) were adjusted for baseline sociodemographics.

**Results** We identified five clusters of SA and DP sequences: (1) 'low or no SA or DP' (88.7% of the population), (2) 'SA due to other than mental diagnosis' (5.2%), (3) 'SA due to mental diagnosis' (3.4%), (4) 'not eligible for SA or DP' (1.4%) and (5) 'DP' (1.2%). Men, highly educated, born outside Sweden and high-income earners were more likely to belong to the first and the fourth cluster (ORs 1.13–4.49). The second, third and fifth clusters consisted mainly of women, low educated and low-income (ORs 1.22–8.90). There were only small differences between branches of industry in adjusted analyses, and many were not significant.

**Conclusion** In general, only a few privately employed white-collar workers had SA and even fewer had DP during the 7-year follow-up. The risk of belonging to a cluster characterised by SA or DP varied by sex, levels of education and income, and other sociodemographic factors.

## INTRODUCTION

Sickness absence (SA) and disability pension (DP) have adverse consequences for individuals, their employers and welfare states. The development of SA and DP is often a long process and varies with type of occupation and work tasks.[1 2] In general, white-collar workers have a lower risk of SA and DP compared with other occupational groups.[3–5] Nevertheless, they constitute a large part of the workforce—approximately half in Sweden in 2018,[6] and about half of them are privately employed.[7] Thus, work incapacity in this group can impose high costs for employees, employers and the welfare state. To prevent work incapacity in this population, more knowledge is needed on the determinants and the process of developing long-term work incapacity.

Previous research on SA and DP within specific occupations or occupational groups has mainly focused on so-called high-risk groups, for example, manual workers and blue-collar workers,[8–14] while studies on white-collar workers are scarce. Those conducted are mainly based on small sample sizes.[15 16] Most of the research on SA and DP among white-collar employees has focused on publicly employed white-collar employees; for instance, the Whitehall-II studies of British civil servants.[17–20] These studies have shown that there are differences in rates of SA among white-collar workers by age,

gender, education, occupational status, and other sociodemographic and socioeconomic factors.

## STRENGTHS AND LIMITATIONS OF THIS STUDY

⇒ Detailed sociodemographic microdata, linked from different population-based registers, about a cohort of all privately employed white-collar workers in Sweden in 2012.
⇒ Long study period (7 years) with no los to follow-up and no bias from self-reports.
⇒ Use of sequence analysis to capture the heterogeneity of the different sickness absence and disability pension patterns over time.
⇒ How many and which states to include in the analyses is dependent on researcher judgement and thus can be arbitrary.

Studies on white-collar workers in the private sector are even more limited. In general, large-scale studies have demonstrated that SA rates in the private sector are generally lower than in the public sector.[21 22] There are several studies on SA and/or DP among private-sector employees, however, hardly any specifically among white-collar workers, despite how many it concerns. Moreover, the few such studies are mainly based on small, selected populations, have large drop-out rates and are mainly based on self-reported data.[5 23–25] So far, only three large-scale studies on private sector white-collar employees have been published: two Swedish studies[3 26] and a Greek study on private sector employees (also including blue-collar employees) that found a smaller SA rate in the shipyard industry than in other industries.[27] The results of the two Swedish studies showed that the risk of SA and DP—and the risk of belonging to an adverse SA/DP trajectory—differed among white-collar workers by age, sex, education, a branch of industry, psychosocial exposures at work and other sociodemographic factors. Further, none of these studies have accounted for transitions between other labour market states in addition to SA and DP, such as employment and unemployment. More studies using full population data with a longitudinal research design are needed to increase the knowledge base.

Moreover, both SA and DP are complex phenomena affected by many factors. Both increase with age, are lower in people with higher education and non-immigrants, and differ by sex; in most occupations, women have higher SA/DP levels than men, hence it is important to include such factors in studies of future SA/DP.[3 28–30]

Sequence analysis is a good method to study developments over time. Unlike more traditionally used methods, such as event history analysis or growth curve models, sequence analysis can describe the duration and frequency of multiple categorical statuses. This holistic perspective is essential in providing an overview of the future development of SA and DP, and in identifying potential sub-groups within a population who share particular patterns in terms of such SA and DP.

The aim of this study was to identify sequences of white-collar workers in the private sector who follow future similar sequences of SA and DP days/year and second, to analyse the sociodemographic and diagnostic composition of the observed clusters of SA and DP.

## METHODS
### Data sources and population
We conducted a 7-year prospective population-based cohort study. We used microdata from the following three nationwide Swedish administrative registers, linked at the individual level by personal identity number (a unique 10-digit number assigned to all Swedish residents)[31]:

– The Longitudinal Integration Database for Health Insurance and Labour Market Studies held by Statistics Sweden, to identify the study cohort and for information on sociodemographic characteristics at baseline 2012 and regarding being in paid work or not in 2012–2018 (see Sociodemographic and work-related variables) or emigrating in 2013–2018.

– The MicroData for Analysis of the Social Insurance database held by the Swedish Social Insurance Agency, for information on SA and DP in the years 2012–2018 (dates, grades (full time or part time) and diagnoses).

– The Cause of Death Register held by the National Board of Health and Welfare for year of death.

The study population consisted of all individuals aged 18–67 years who lived in Sweden on both 31 December 2011 and 31 December 2012, who had an occupational code according to the Swedish Standard for Occupational Classification (SNI) indicating a white-collar occupation,[3] were employed at a private-sector company during 2012, and had an income from work, parental benefits, SA and/or DP that amounted to at least 75% of the necessary income level to qualify for SA benefits from the Social Insurance Agency (SEK7920 in 2012, approximately €910 by the 2012 exchange rate, updated yearly in line with inflation). We excluded unemployed, self-employed, and those who were on full-time DP for the entire year 2012 (n=461). The total study cohort included 1 283 516 individuals.

### Public SA insurance in Sweden
In Sweden, all residents aged at least 16 years with an income from work or unemployment benefits who have a reduced work capacity due to morbidity are covered by the national public SA insurance.[32] A physician's certificate is required after 7 days. After an unpaid qualifying day, the employer pays the following 13 SA days, after which SA benefits are paid by the Social Insurance Agency. For the unemployed, the Social Insurance Agency pays after the first qualifying day. Thus, we excluded SA spells shorter than 15 days, in order not to introduce bias, since we only had information of SA spells exceeding 14 days for the employed. There was no limitation regarding how long an SA spell could be ongoing for. Residents in Sweden aged 19–64 years, whose work capacity is long-term or permanently reduced, can be granted DP from the Social Insurance Agency. SA covers about 80% and DP about 65% of lost income, both up to a certain level. Both SA and DP can be granted for part-time or full-time (25%, 50%, 75% or 100% of ordinary work hours). This means that people can be on partial SA and DP at the same time.

### Sociodemographic and work-related variables
We included information on sex, age group, country of birth, educational level, family composition, type of living area and branch of industry based on the SNI categorised into the following six groups: manufacturing, services, transport, construction and installation, care and education, or commerce and hospitality. All variables were measured at the baseline year 2012.

## Measures on SA and DP

We used SA net days/year and DP net days/year as outcomes. Net days were calculated so that partial days of SA or DP were combined, for example, 2 days of part-time SA for 50% were summed to one net day, and a similar procedure was used for DP days. The first 14 days of SA spells (>14 days) were counted as being of the same grade as day 15 for the purpose of calculating net days. The number of SA net days in 2012 were categorised as shown in table 1. The SA diagnoses were categorised into the following seven International Classification of Disease groups[33]: cancer (C00–D48), mental diagnoses (F00–F99 and Z73), circulatory diseases (I00–I99), musculoskeletal diagnoses (M00–M99), pregnancy-related diagnoses (O00–O99), injuries (S00–T98) and other diagnostic groups (all others, including missing diagnosis (approximately 1% of all spells). In the multinomial logistic regression, pregnancy-related diagnoses were dropped, as no men could have pregnancy-related diagnoses, which made it highly correlated with sex.

In analyses of the yearly states of SA/DP, all diagnoses other than mental and musculoskeletal diseases were combined to form one status. Any DP, regardless of diagnosis, was considered as one group.

## Sequence analysis and multinomial regression analysis

We used sequence analysis to examine different statuses of SA and DP days/year, and the transitions between such statuses. SA and DP status was measured on a yearly basis for each of the seven follow-up years and was coded into one of the following seven statuses:

1. No SA or DP.
2. SA due to mental diagnoses but no DP.
3. SA due to musculoskeletal diagnoses but no DP.
4. SA due to other diagnoses but no DP.
5. Both SA and DP.
6. Only DP.
7. Ineligible for SA and DP (due to being emigrated, dead, retired, or having no qualifying income from work or work-related benefits).

Individuals who had SA in more than one diagnostic category were assigned to the diagnostic category they had the most days in that year. We illustrated the individual and proportional changes in SA/DP statuses over time with sequence index plots and status proportion plots.[34]

We used an optimal matching (OM) method to group similar sequences with each other. OM measures the dissimilarities through the changes needed to make two sequences identical.[35] In other words, the OM algorithm creates metric distances between two sequences, which can be defined as the minimum combination of replacements, insertion and deletions to transform one sequence to another.[36] We used R statistical program version V.4.1.0 and packages TraMineR and nnet for the sequence analysis.

We used multinomial regression analysis to analyse how sociodemographic characteristics and branch of industry were associated with each of the obtained clusters, using the first cluster as the reference category. ORs with their 95% CIs were reported.

**Table 1** Characteristics of the study cohort in 2012

| | Total | |
|---|---|---|
| | n | % |
| **Sex** | | |
| Women | 598 965 | 47.59 |
| Men | 659 755 | 52.41 |
| **Age group** | | |
| 18–24 | 63 788 | 5.07 |
| 25–34 | 271 754 | 21.59 |
| 35–44 | 371 803 | 29.54 |
| 45–54 | 322 900 | 25.65 |
| 55–64 | 117 802 | 9.36 |
| 65–67 | 110 673 | 8.79 |
| **Type of living area** | | |
| Large city | 647 868 | 51.47 |
| Medium-sized town | 384 746 | 30.57 |
| Rural or small town | 226 106 | 17.96 |
| **Educational level** | | |
| Primary | 61 256 | 4.87 |
| Secondary | 521 351 | 41.42 |
| Tertiary | 676 113 | 53.71 |
| **Country of birth** | | |
| Sweden | 1 129 201 | 89.71 |
| Other Nordic country | 26 478 | 2.10 |
| Other EU25 country‡ | 25 010 | 1.99 |
| Other countries | 78 031 | 6.20 |
| **Family composition** | | |
| Couple without children <18 at home | 167 791 | 13.33 |
| Couple with children <18 at home | 595 073 | 47.28 |
| Single without children <18 at home | 411 846 | 32.72 |
| Single with children <18 at home | 84 010 | 6.67 |
| **Branch of industry** | | |
| Manufacturing | 259 419 | 20.61 |
| Service | 543 452 | 43.17 |
| Trade, hotel, restaurant | 161 308 | 12.82 |
| Transport | 54 978 | 4.37 |
| Construction | 49 938 | 3.97 |
| Education, care, nursing, social services | 189 083 | 15.02 |
| Unknown | 542 | 0.04 |
| **Income (SEK)** | | |
| SEK7920–SEK87 999 | 23 701 | 1.88 |
| SEK88 000–SEK175 999 | 81 257 | 6.46 |

Continued

| Table 1 Continued | | |
|---|---|---|
| | Total | |
| | n | % |
| SEK176 000–SEK329 999 | 355 583 | 28.25 |
| SEK330 000–SEK439 999 | 347 772 | 27.63 |
| >SEK440 000 | 450 407 | 35.78 |
| No of SA net days in 2012 in SA spells >14 gross days | | |
| 0 | 1 170 169 | 92.96 |
| 1–14 | 27 895 | 2.22 |
| 15–30 | 17 001 | 1.35 |
| 31–90 | 24 292 | 1.93 |
| 91–180 | 10 885 | 0.86 |
| 181–365 | 7405 | 0.59 |
| 366* | 1071 | 0.09 |
| Total | 1 258 720 | 100.00 |
| SA diagnoses in 2012† | | |
| Mental diagnoses | 27 765 | 2.21 |
| Musculoskeletal diagnoses | 18 502 | 1.44 |
| Injury | 9179 | 0.72 |
| Cancer | 5294 | 0.41 |
| Circulatory diagnoses | 3884 | 0.30 |
| Pregnancy-related diagnoses | 7005 | 0.55 |
| Other diagnoses | 23 539 | 1.83 |

*2012 was a leap year.
†Individuals could have had several SA spells with different diagnoses.
‡Refers to the 25 countries of the European Union in 2004-2007
SA, sickness absence; SEK, Swedish Krona.

## Patient and public involvement

Representatives from the private white-collar sector in Sweden, both for employees and employers (the labour union PTK, the Confederation of Swedish Enterprise and Alecta) were involved in selecting the research questions through joint meetings throughout the project period, and afterwards in disseminating results.

## RESULTS

### Characteristics of the study population

Table 1 shows the characteristics of the study cohort of the 1 283 516 privately employed white-collar workers. There were slightly more men (52.4%) in the cohort. The largest age group was those aged 35–44 years (29.5%), over half lived in a large city (51.5%) and had a tertiary education (53.7%). The majority were born in Sweden (89.7%), and almost half were married or cohabiting and having children below the age of 18 at home (47.3%). The largest group was the service industry (43.1%) and the largest income group was those who earned over

SEK440 000 (around €50 556 according to the average 2012 conversion rate) per year (35.8%). A large majority did not have any SA in 2012: only around 7% had at least one SA spell >14 days. Around 2.2% had SA due to mental diagnoses, 1.4% due to musculoskeletal diagnoses and around 3.8% due to any other diagnoses.

### Clusters of SA and DP trajectories

We identified five different groups of sequences, that is, clusters. Figure 1 shows each of the five clusters, as well as the proportion of individuals in each cluster and the proportion of individuals within the respective clusters in each state during each year. The sociodemographic characteristics of each cluster can be seen in online supplemental table 1. The first cluster (n=1 138 777, 88.7% of all in the cohort) was the largest one, and almost 95% of individuals in this cluster had no SA or DP days. We called this cluster 'low or no SA or DP'.

Cluster 2 (n=66 997, 5.2%), which was the second largest, was characterised by SA due other than mental diagnosis, including those with mainly musculoskeletal diagnoses (figure 1). We called this cluster 'SA due to other diagnoses'.

Cluster 3 (n=43 871, 3.4%) consisted mostly of those who had SA mainly due to mental diagnoses (figure 1). We called this cluster 'SA due to mental diagnoses'.

Cluster 4 (n=18 150, 1.4%) was characterised by individuals who were not eligible for SA or DP since they either died, emigrated or left the labour force (figure 1). We called this cluster 'ineligible for SA and DP'.

The smallest cluster, cluster 5 (n=15 721, 1.2%) was characterised by individuals who had either partial or full-time DP (figure 1). We called this cluster 'DP'.

To better understand the most common SA and DP sequences, we examined the 20 most frequent sequences (online supplemental figure 1). Most (68.4%) had no SA or DP during the follow-up. The remaining trajectories largely consisted of sequences where individuals had SA for 1 year and then returned to no SA or DP. Very few had DP during the follow-up.

### The associations between individual characteristics and belonging to clusters of SA and DP

To study how individual characteristics and SA at baseline were associated with cluster membership, we used multinomial regression analysis. cluster 1 'low or no SA or DP' was used as the reference category since it was the largest and most homogeneous in its sequence content (table 2). Cluster 1 could be described as consisting of men of younger working-age, who had high levels of education and income, worked in service industry or in manufacturing and had no or only little SA in 2012 (online supplemental table 1).

In the fully adjusted models, compared with cluster 1 'low or no SA or DP', women (men having an OR of 0.47 (95% CI 0.46 to 0.47)), over or under 35–44 years (but not over 64 years), those with less than tertiary education, who were born outside EU25 countries (i.e., the 25

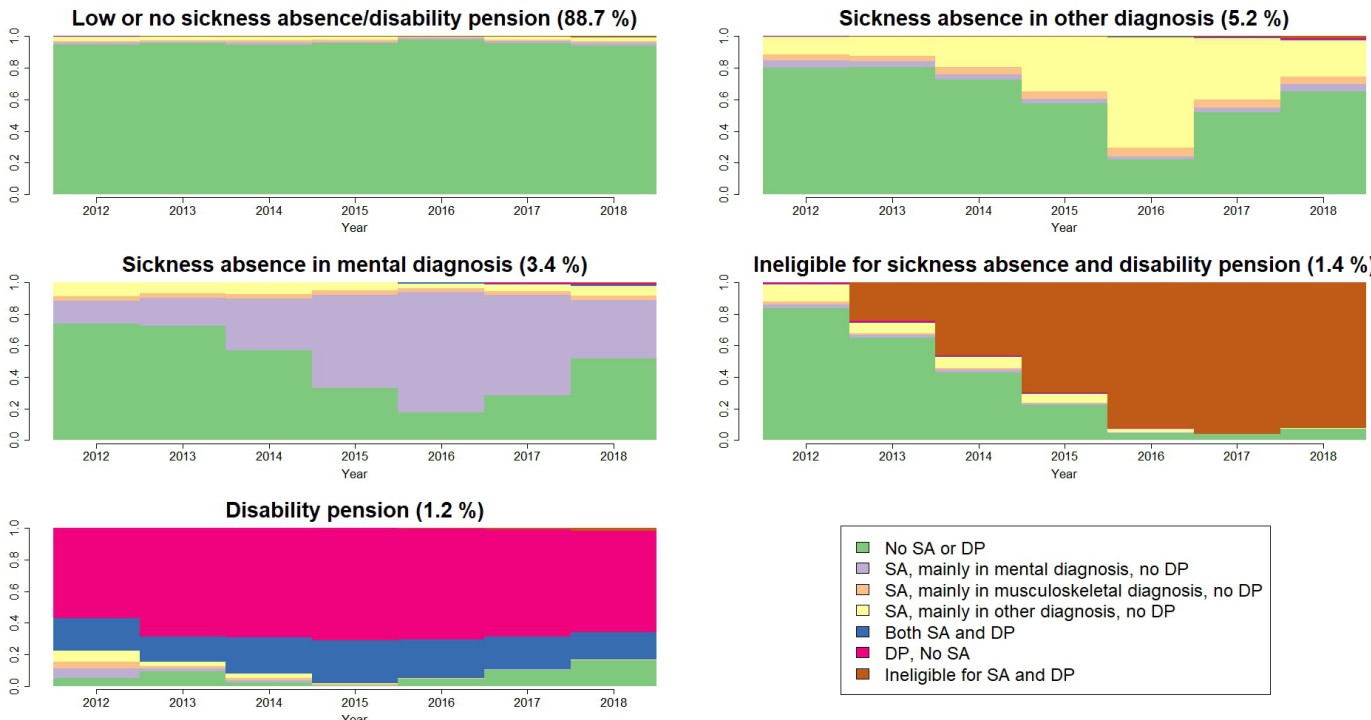

**Figure 1** Density plot of sickness absence (SA) and disability pension (DP) visualising the proportion of each SA and DP status for each cluster over the follow-up.

countries that made up the European Union from 2004 to 2007), living with children, worked in service industry or education, care, nursing or social service industry, had a medium income, had any SA and especially SA due to cancer or due to 'other diagnoses' were more likely to belong to cluster 2 'SA due to mental diagnoses' (table 2).

Women (men having an OR 0.38 (95% CI 0.37 to 0.39)), 34–44 years, who had less than tertiary education, who were single living with children, worked in education, care, nursing or social service industry, had medium low income, had more than 180 SA days in 2012, especially due to mental diagnoses, had the highest ORs of belonging to cluster 3 'SA due to mental diagnoses' (table 2). The second and third clusters could be described as consisting of working-age women, who had less than tertiary education and medium income, who worked in education, care, nursing or social service industry and had some SA in baseline year, especially due to mental diagnoses in the third cluster (online supplemental table 1).

The OR for belonging to cluster 4 'ineligible for SA and DP' was the highest in men (OR 1.13; 95% CI 1.10 to 1.17), 65–67 year, had primary education, lived without children, were born outside Sweden, had a very low income, who worked in trade, hotel or restaurant industry or transport industry, had >180 SA days in 2012 and had SA due to circulatory diagnoses (table 2). The fourth cluster could be described as consisting of men over 64 years, who had primary education and were born outside Sweden, had low income and had long-term SA in 2012, especially due to cancer (online supplemental table 1).

The OR of belonging to cluster 5 'DP' were higher in women (OR 0.69 (95% CI 0.66 to 0.72) in men), 45–64

years, who had less than tertiary education, were born in Sweden, who were single, worked in manufacturing, had low to medium low income, had at least 30 SA days in 2012 and especially those with SA due to circulatory diagnoses (table 2). This fifth cluster could be described as consisting of older working age women, with low education, working in service industry with low income and long-term SA at baseline (online supplemental table 1).

## Discussion

In this large prospective cohort study of all 1.3 million privately employed white-collar workers in Sweden in 2012, we analysed the development of their future number of SA and DP days/year up through 2018. In general, most of the employees had no SA during the follow-up and DP was even rarer. We found five clusters of future SA and DP trajectories: (1) 'low or no SA or DP' (88.7% of all), (2) 'SA due to other (than mental) diagnosis' (5.2%), (3) 'SA due to mental diagnosis' (3.4%), (4) 'not eligible for SA or DP' (1.4%) and (5) 'DP' (1.2%). These results suggest that the majority of privately employed white-collar workers were doing well in terms of SA/DP.

We found some differences related to sociodemographic factors in terms of belonging to different sequence clusters. Many of those in cluster 1 'low or no SA or DP' were Swedish-born, 25–54 years, highly educated, and high-income earning men, who lived in a large city, and were married or cohabiting with children at home. The same sociodemographic characteristics are typically associated with lower risk of SA or DP in longitudinal nationwide studies.[28 29]

**Table 2** Multinomial regression with five clusters of sickness absence (SA) and disability pension (DP) days/year among privately employed white-collar workers, ORs with their 95% CIs, cluster 1 'low or no SA or DP' was used as reference group

| | Cluster 2 SA due to other diagnoses | | Cluster 3 SA due to mental diagnoses | | Cluster 4 Ineligible for SA and DP | | Cluster 5 DP | |
|---|---|---|---|---|---|---|---|---|
| | Crude OR | Adjusted OR | Crude OR | Adjusted OR | Crude OR | Adjusted OR | Crude OR | Adjusted OR |
| **Sex** | | | | | | | | |
| Women | ref. | ref. | ref. | ref. | ref. | ref. | ref. | ref. |
| Men | 0.38 (0.37 to 0.38) | 0.47 (0.46 to 0.47) | 0.30 (0.29 to 0.31) | 0.38 (0.37 to 0.39) | 1.18 (1.15 to 1.22) | 1.13 (1.10 to 1.17) | 0.32 (0.31 to 0.33) | 0.69 (0.66 to 0.72) |
| **Age group** | | | | | | | | |
| 18–24 | 1.36 (1.31 to 1.41) | 1.09 (1.05 to 1.14) | 0.76 (0.73 to 0.80) | 0.37 (0.35 to 0.39) | 1.34 (1.25 to 1.44) | 1.54 (1.42 to 1.66) | 0.28 (0.24 to 0.34) | 0.00 (0.00 to 0.00) |
| 25–34 | 1.33 (1.30 to 1.36) | 1.30 (1.27 to 1.33) | 0.96 (0.94 to 0.99) | 0.79 (0.77 to 0.81) | 1.34 (1.28 to 1.4) | 1.54 (1.47 to 1.61) | 0.30 (0.27 to 0.33) | 0.10 (0.09 to 0.11) |
| 35–44 | ref. | ref. | ref. | ref. | ref. | ref. | ref. | ref. |
| 45–54 | 1.37 (1.34 to 1.40) | 1.42 (1.39 to 1.45) | 0.88 (0.86 to 0.90) | 0.79 (0.77 to 0.82) | 1.12 (1.07 to 1.17) | 1.28 (1.22 to 1.34) | 2.61 (2.50 to 2.74) | 3.34 (3.18 to 3.51) |
| 55–64 | 1.74 (1.70 to 1.79) | 1.73 (1.67 to 1.78) | 0.71 (0.68 to 0.74) | 0.57 (0.54 to 0.59) | 1.94 (1.85 to 2.05) | 1.99 (1.88 to 2.10) | 4.93 (4.69 to 5.18) | 4.29 (4.04 to 4.55) |
| 65–67 | 0.50 (0.48 to 0.52) | 0.45 (0.43 to 0.47) | 0.12 (0.11 to 0.13) | 0.10 (0.09 to 0.11) | 2.50 (2.38 to 2.62) | 2.32 (2.20 to 2.45) | 2.68 (2.53 to 2.83) | 1.26 (1.18 to 1.35) |
| **Type of living area** | | | | | | | | |
| Large city | ref. | ref. | ref. | ref. | ref. | ref. | ref. | ref. |
| Medium-sized town | 1.02 (1.00 to 1.04) | 1.02 (1.00 to 1.04) | 0.99 (0.97 to 1.02) | 1.06 (1.03 to 1.08) | 0.75 (0.72 to 0.77) | 0.82 (0.79 to 0.85) | 1.70 (1.64 to 1.76) | 1.05 (1.01 to 1.10) |
| Rural or small town | 1.11 (1.08 to 1.13) | 1.07 (1.05 to 1.09) | 1.04 (1.01 to 1.07) | 0.97 (0.94 to 0.99) | 0.71 (0.68 to 0.74) | 0.67 (0.64 to 0.70) | 2.44 (2.34 to 2.54) | 1.03 (0.99 to 1.08) |
| **Educational level** | | | | | | | | |
| Primary | 1.37 (1.32 to 1.42) | 1.66 (1.60 to 1.73) | 1.20 (1.15 to 1.26) | 1.79 (1.71 to 1.87) | 1.57 (1.48 to 1.66) | 1.39 (1.32 to 1.48) | 4.12 (3.89 to 4.36) | 1.68 (1.57 to 1.79) |
| Secondary | 1.41 (1.38 to 1.43) | 1.34 (1.32 to 1.36) | 1.25 (1.23 to 1.28) | 1.22 (1.19 to 1.24) | 0.78 (0.76 to 0.81) | 0.86 (0.84 to 0.89) | 2.68 (2.59 to 2.78) | 1.50 (1.44 to 1.56) |
| Tertiary | ref. | ref. | ref. | ref. | ref. | ref. | ref. | ref. |
| **Country of birth** | | | | | | | | |
| Sweden | ref. | ref. | ref. | ref. | ref. | ref. | ref. | ref. |
| Other Nordic country | 1.21 (1.15 to 1.27) | 1.04 (0.99 to 1.10) | 1.13 (1.06 to 1.20) | 0.86 (0.80 to 0.92) | 3.90 (3.67 to 4.14) | 3.25 (3.05 to 3.46) | 1.69 (1.55 to 1.85) | 0.80 (0.72 to 0.89) |
| Other EU25 country† | 1.00 (0.94 to 1.06) | 1.07 (1.01 to 1.13) | 1.00 (0.93 to 1.07) | 1.11 (1.05 to 1.19) | 4.03 (3.79 to 4.27) | 4.49 (4.25 to 4.76) | 0.86 (0.76 to 0.97) | 0.31 (0.26 to 0.37) |
| Other countries | 1.41 (1.37 to 1.45) | 1.13 (1.10 to 1.17) | 1.16 (1.12 to 1.21) | 0.87 (0.84 to 0.91) | 2.30 (2.20 to 2.41) | 2.58 (2.46 to 2.70) | 0.87 (0.81 to 0.93) | 0.48 (0.44 to 0.52) |
| **Family composition** | | | | | | | | |
| Couple without children <18 at home | ref. | ref. | ref. | ref. | ref. | ref. | ref. | ref. |
| Couple with children <18 at home | 0.94 (0.92 to 0.97) | 0.98 (0.95 to 1.00) | 1.65 (1.59 to 1.71) | 0.79 (0.76 to 0.82) | 0.52 (0.50 to 0.54) | 0.62 (0.59 to 0.65) | 0.36 (0.34 to 0.37) | 0.71 (0.67 to 0.74) |

Continued

**Table 2** Continued

| | Cluster 2 SA due to other diagnoses | | Cluster 3 SA due to mental diagnoses | | Cluster 4 Ineligible for SA and DP | | Cluster 5 DP | |
|---|---|---|---|---|---|---|---|---|
| | Crude OR | Adjusted OR | Crude OR | Adjusted OR | Crude OR | Adjusted OR | Crude OR | Adjusted OR |
| Single without children <18 at home | 1.13 (1.10 to 1.16) | 1.19 (1.15 to 1.22) | 1.71 (1.65 to 1.78) | 0.93 (0.90 to 0.97) | 0.97 (0.93 to 1.01) | 1.04 (0.99 to 1.08) | 0.53 (0.50 to 0.55) | 1.22 (1.16 to 1.28) |
| Single with children <18 at home | 1.69 (1.63 to 1.74) | 1.35 (1.30 to 1.40) | 3.68 (3.52 to 3.84) | 1.31 (1.25 to 1.36) | 0.61 (0.57 to 0.66) | 0.54 (0.50 to 0.59) | 0.87 (0.82 to 0.92) | 1.05 (0.98 to 1.13) |
| Branch of industry | | | | | | | | |
| Manufacturing | 0.82 (0.80 to 0.84) | 0.89 (0.87 to 0.91) | 0.69 (0.67 to 0.71) | 0.83 (0.80 to 0.85) | 1.01 (0.97 to 1.05) | 1.07 (1.03 to 1.11) | 0.63 (0.60 to 0.66) | 1.11 (1.05 to 1.17) |
| Service | ref. | ref. | ref. | ref. | ref. | ref. | ref. | ref. |
| Trade, hotel, restaurant | 1.11 (1.08 to 1.13) | 0.89 (0.87 to 0.91) | 1.04 (1.01 to 1.07) | 0.92 (0.89 to 0.95) | 0.89 (0.85 to 0.94) | 1.10 (1.05 to 1.16) | 0.93 (0.89 to 0.98) | 0.67 (0.63 to 0.71) |
| Transport | 1.19 (1.14 to 1.23) | 1.02 (0.98 to 1.07) | 0.91 (0.87 to 0.96) | 0.85 (0.81 to 0.90) | 1.01 (0.94 to 1.09) | 1.11 (1.03 to 1.19) | 1.08 (1.00 to 1.16) | 0.66 (0.60 to 0.72) |
| Construction | 0.92 (0.88 to 0.96) | 1.01 (0.97 to 1.06) | 0.67 (0.63 to 0.71) | 0.45 (0.42 to 0.49) | 0.71 (0.65 to 0.77) | 0.98 (0.90 to 1.06) | 0.95 (0.88 to 1.04) | 0.78 (0.71 to 0.86) |
| Education, care, nursing, social services | 2.03 (1.99 to 2.07) | 1.34 (1.31 to 1.37) | 1.82 (1.77 to 1.86) | 1.19 (1.16 to 1.22) | 0.98 (0.94 to 1.03) | 1.00 (0.96 to 1.05) | 1.73 (1.66 to 1.80) | 0.80 (0.76 to 0.83) |
| Income (SEK) | | | | | | | | |
| SEK7920–SEK87 999 | 0.68 (0.64 to 0.72) | 0.62 (0.58 to 0.66) | 0.66 (0.62 to 0.71) | 0.75 (0.70 to 0.81) | 4.25 (3.99 to 4.52) | 3.81 (3.55 to 4.10) | 1.80 (1.67 to 1.94) | 4.50 (4.17 to 4.87) |
| SEK88 000–SEK175 999 | 0.90 (0.87 to 0.93) | 0.94 (0.91 to 0.98) | 1.00 (0.96 to 1.03) | 1.17 (1.13 to 1.21) | 1.37 (1.29 to 1.46) | 1.06 (0.99 to 1.14) | 4.36 (4.21 to 4.52) | 8.90 (8.57 to 9.25) |
| SEK176 000–SEK329 999 | ref. | ref. | ref. | ref. | ref. | ref. | ref. | ref. |
| SEK330 000–SEK439 999 | 0.67 (0.66 to 0.69) | 0.89 (0.87 to 0.91) | 0.57 (0.55 to 0.58) | 0.73 (0.71 to 0.74) | 0.81 (0.77 to 0.84) | 1.20 (1.15 to 1.25) | 0.16 (0.15 to 0.17) | 0.12 (0.11 to 0.12) |
| >SEK440000 | 0.39 (0.38 to 0.40) | 0.58 (0.57 to 0.60) | 0.30 (0.29 to 0.31) | 0.45 (0.44 to 0.47) | 1.03 (0.99 to 1.07) | 1.56 (1.49 to 1.63) | 0.05 (0.05 to 0.06) | 0.00 (0.00—0.00) |
| No of SA net days in 2012 | | | | | | | | |
| 0 | ref. | ref. | ref. | ref. | ref. | ref. | ref. | ref. |
| 1–14 | 3.89 (3.76 to 4.03) | 3.59 (3.47 to 3.72) | 4.47 (4.29 to 4.66) | 2.91 (2.77 to 3.05) | 1.60 (1.45 to 1.75) | 1.72 (1.56 to 1.89) | 5.71 (5.35 to 6.09) | 3.36 (3.10 to 3.65) |
| 15–30 | 4.17 (3.99 to 4.35) | 2.80 (2.67 to 2.94) | 4.97 (4.73 to 5.23) | 4.04 (3.84 to 4.26) | 2.10 (1.89 to 2.33) | 1.22 (1.06 to 1.41) | 5.48 (5.04 to 5.96) | 3.27 (2.95 to 3.62) |
| 31–90 | 4.56 (4.40 to 4.73) | 4.10 (3.95 to 4.25) | 6.50 (6.25 to 6.76) | 4.70 (4.50 to 4.90) | 2.96 (2.74 to 3.20) | 2.44 (2.23 to 2.67) | 8.36 (7.87 to 8.89) | 6.78 (6.34 to 7.26) |
| 91–180 | 5.62 (5.32 to 5.93) | 5.42 (5.14 to 5.72) | 9.92 (9.40 to 10.46) | 7.71 (7.28 to 8.17) | 5.92 (5.4 to 6.49) | 5.11 (4.60 to 5.67) | 22.85 (21.45 to 24.35) | 20.67 (19.25 to 22.2) |
| 181–365 | 6.43 (6.00 to 6.89) | 4.73 (4.41 to 5.08) | 16.10 (15.16 to 17.10) | 10.92 (10.25 to 11.63) | 13.88 (12.74 to 15.11) | 13.60 (12.42 to 14.89) | 47.31 (44.34 to 50.49) | 22.75 (21.08 to 24.55) |
| 366* | 7.86 (6.45 to 9.57) | 2.65 (2.07 to 3.39) | 27.08 (23.16 to 31.67) | 4.76 (3.78 to 5.98) | 36.19 (30.22 to 43.33) | 50.60 (42.93 to 59.65) | 100.07 (85.89 to 116.59) | 61.44 (52.82 to 71.47) |
| SA diagnoses in 2012 | | | | | | | | |

Continued

**Table 2** Continued

| | Cluster 2 SA due to other diagnoses | | Cluster 3 SA due to mental diagnoses | | Cluster 4 Ineligible for SA and DP | | Cluster 5 DP | |
|---|---|---|---|---|---|---|---|---|
| | Crude OR | Adjusted OR | Crude OR | Adjusted OR | Crude OR | Adjusted OR | Crude OR | Adjusted OR |
| Mental diagnoses | 3.31 (3.18 to 3.44) | 2.80 (2.69 to 2.91) | 11.74 (11.39 to 12.10) | 8.27 (8.00 to 8.54) | 1.87 (1.71 to 2.05) | 2.44 (2.24 to 2.65) | 8.76 (8.32 to 9.22) | 6.71 (6.32 to 7.12) |
| Musculoskeletal diagnoses | 4.02 (3.86 to 4.19) | 3.16 (3.03 to 3.30) | 3.06 (2.90 to 3.24) | 2.47 (2.33 to 2.62) | 1.88 (1.69 to 2.08) | 1.54 (1.37 to 1.73) | 10.54 (9.98 to 11.13) | 6.63 (6.23 to 7.06) |
| Injury | 3.31 (3.12 to 3.52) | 3.09 (2.91 to 3.29) | 2.35 (2.16 to 2.55) | 2.43 (2.24 to 2.64) | 1.94 (1.69 to 2.23) | 2.66 (2.35 to 3.02) | 4.95 (4.49 to 5.45) | 3.23 (2.88 to 3.61) |
| Cancer | 4.02 (3.71 to 4.37) | 4.04 (3.73 to 4.37) | 2.09 (1.83 to 2.38) | 1.80 (1.57 to 2.06) | 23.29 (21.69 to 25.01) | 19.35 (17.86 to 20.97) | 4.83 (4.19 to 5.58) | 3.04 (2.61 to 3.55) |
| Circulatory diagnoses | 3.14 (2.85 to 3.46) | 3.32 (2.97 to 3.71) | 1.99 (1.72 to 2.30) | 2.30 (1.94 to 2.72) | 3.60 (3.06 to 4.25) | 6.61 (5.84 to 7.48) | 11.96 (10.78 to 13.27) | 18.02 (16.09 to 20.19) |
| Other diagnoses | 5.09 (4.92 to 5.28) | 3.93 (3.79 to 4.08) | 4.29 (4.10 to 4.49) | 3.56 (3.40 to 3.73) | 2.79 (2.57 to 3.03) | 1.25 (1.11 to 1.41) | 9.67 (9.18 to 10.19) | 7.25 (6.82 to 7.69) |

*2012 was a leap year, thus those individuals were on full-time SA the whole year.
†Refers to the 25 countries of the European Union in 2004-2007

We also found that female sex, low education, low income, and working in education, care, nursing, or social services were associated with a higher risk of belonging to clusters characterised by at least some SA or DP. Similar results were found in a previous cross-sectional study using the same data with number and prevalence of SA days as outcomes,[3] as well as studies on SA and DP among white-collar workers in the retail and wholesale industry.[26 37] In general, previous longitudinal population-based studies have consistently found that women, low educated and low-and income earners,[28 29] and those working in health-care and service industries[22] have a higher risk of SA and/or DP. While these characteristics—low education, low income and working in the healthcare industry—are usually considered as explanations to why blue-collar workers have a higher risk of SA or DP than white-collar workers,[4 38] our results indicate that the same risk factors apply within white-collar employees working in the private sector. More knowledge is warranted regarding potential mechanisms behind this.

It is understandable that SA due to mental diagnoses constituted an independent cluster since among white-collar workers that is the most common specific diag-nostic group of SA and/or DP.[1 37 39–41] This cluster was more common among women, 34–44 years, less than tertiary educated, low-income earners who worked in education, care, nursing and social industry, and had a long SA spell in 2012, which are known risk factors for SA due to mental diagnoses in general.[42 43]

The cluster 'ineligible for SA or DP' had relatively many individuals aged ≥ 55 years, which makes sense since those who left paid work (eg, through old-age pension) or died during the follow-up belonged to this cluster. There were also many highly educated and high-income earners, who typically are occupationally and geographically mobile, in this cluster. Relatively many of them were born outside Sweden; hence many of them probably emigrated from Sweden. Those who had SA due to cancer in 2012, had higher OR of belonging to this cluster than to any other cluster.

We found that the estimates for associations between branch of industry and cluster attenuated in the adjusted analyses, indicating that differences between the various branches of industry were more related to other factors. The Swedish Social Insurance Agency has found that in Sweden, occupation is more closely associated with SA than branch of industry.[44] However, to what extent this is true within the group white-collar workers is unknown and should be further studied.

### Strength and limitations

Strengths of this study are the use of a large, population-based cohort the use of linked microdata from three high-quality nationwide registers without dropouts, the long prospective follow-up, and that all data were administrative, not self-reports with possible bias. Using sequence analysis allowed us to

explore specific subgroups in the development of SA and DP. Other strengths are that all included were covered by the same public SA and DP insurances, and the high employment-frequency in Sweden, that is, the healthy-worker effect did not bias the result much.

Since the study population consisted of privately employed white-collar workers in Sweden, the results cannot directly be generalised to other types of occupational populations or to other countries with other SA/DP systems or employment frequencies. Future studies might choose to explore other, or more specific SA states, regarding number of SA days or part-time and full-time SA/DP. As this was an observational study, no causal inferences can be drawn from the results.

## CONCLUSION

In general, privately employed white-collar workers rarely had SA and even more rarely DP days during the 7-year follow-up. The risk of belonging to a cluster characterised by receiving SA varied by sex, levels of education and income, branch of industry and other sociodemographic factors.

**Contributors** KF and KA planned and designed the study. KF supervised the analyses. LS wrote the first draft of the paper. All authors critically revised the paper for intellectual content. All authors approved the submission of the study. KF acts as guarantor for the study.

**Funding** The study was funded by Alecta Insurance, award/grant number is not applicable. We used data from the REWHARD consortium supported by the Swedish Research Council (grant no. 2017-00624).

**Competing interests** None declared.

**Patient and public involvement** Patients and/or the public were involved in the design, or conduct, or reporting, or dissemination plans of this research. Refer to the Methods section for further details.

**Patient consent for publication** Not applicable.

**Provenance and peer review** Not commissioned; externally peer reviewed.

**Data availability statement** Data may be obtained from a third party and are not publicly available. The used data cannot be made publicly available due to privacy regulations. According to the General Data Protection Regulation, the Swedish law SFS 2018:218, the Swedish Data Protection Act, the Swedish Ethical Review Act and the Public Access to Information and Secrecy Act, these types of sensitive data can only be made available for specific purposes that meets the criteria for access to this type of sensitive and confidential data as determined by a legal review. KA ( Kristina.alexanderson@ki.se) can be contacted regarding the data.

**ORCID iD**
Kristin Farrants http://orcid.org/0000-0001-9595-6627

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
