## [Reviewer comments · BMJ Open]

ARTICLE DETAILS

TITLE (PROVISIONAL)	Sequence analysis of sickness absence and disability pension days in 2012–2018 among privately employed white-collar workers in Sweden: a prospective cohort study
AUTHORS	Salonen, Laura; Alexanderson, Kristina; Farrants, Kristin

VERSION 1 – REVIEW

REVIEWER	Maric, Nada University of Banja Luka
REVIEW RETURNED	04-Oct-2023

GENERAL COMMENTS	First I would like to congratulate you on your excellent work. The Introduction is clearly written and leads to the core of the problem. The method is well described. The results are presented clearly. Discussion is good, you compare your results with other studies but I have only one suggestion. In discussion, you could more discuss about what could be a reason for these differences in sociodemographic characteristics between clusters and you could add implications for policymakers.
--

REVIEWER	Mustard, Cameron Institute for Work and Health
REVIEW RETURNED	11-Oct-2023

GENERAL COMMENTS	Sequence analysis of sickness absence and disability pension days in 2012-2018 among privately employed white-collar workers in Sweden: a prospective cohort study. This manuscript reports on an analysis of approximately 1,200,000 male and female white collar workers employed in the Swedish private sector. Data for the analysis was obtained from three administrative data sources. The study design is longitudinal, classifying employment status at baseline in 2012, and following workers over seven years to ascertain the incidence and duration of episodes of sickness absence (greater than 14 days) and the incidence of disability pension. A method termed 'sequence analysis' was used to derive five clusters of workers, and multinomial regression analysis was used to identify sociodemographic characteristics associated with cluster membership. Generally, the manuscript was well structured and frequently clearly written. The study rationale is expressed in the introduction, proposing that previous research on the incidence of sickness absence and disability pensions among white collar workers was limited.
---

Additionally, the authors argue that studies of white collar workers in private sector employment are also not extensive. While I am not persuaded that the authors' assessment that studies of white collar workers was limited is correct, I accept their assessment that there are limited longitudinal cohort studies of white collar workers in private sector employment. That said, I do think the introduction needs to make a stronger case for why a focus on private sector employment is justified.

The description of the sequence analysis methods is relatively brief. I would note that the potentially relevant information presented in Figure 1 (density plots over the seven year followup) was not interpreted for the reader. If this Figure is to be retained, please provide interpretation.

I was also puzzled that the authors did not consider evaluating cluster membership based on the cumulative burden of sickness absence. Specifically, in Cluster 2 and Cluster 3 it appeared to me that there would be value in distinguishing between low incidence/low days and high incidence / high days of sickness absence.

I was curious about the reliability of the SA measure adopted in this study, specifically physician-certified sickness absence episodes of more than 14 days that were paid by the Social Insurance Agency. Can the authors provide information on the proportion of SA episodes potentially eligible for coverage that are not reported to the Social Insurance Agency (and therefore not included in the MiDAS database)? I would also recommend additional details be included to clarify the source(s) of information underlying the status category 'ineligible for SA and DP'. My reading of the manuscript's methods suggests that this status category was derived by the authors from information in the LISA database, rather than an administrative determination in the MiDAS database.

I found the definition of a white collar occupation to be insufficient. Readers are provided the following information: 'an occupational code according to the Swedish Standard for Occupational Classification indicating a white collar occupation'. No citation was provided. Additionally, the manuscript lacked a conceptual definition of a white collar occupation and did not, in my view, adequately describe how employment/working conditions may differ between men and women in this broad occupational category.

On the basis of the information provided in the manuscript's method section, I understand that private sector employment status was defined in 2012 as a baseline characteristic. If I am correct in this assumption, how is change from private to public sector employment addressed. In most high income countries, the annual incidence of job change (either through job leaving or job losing) is in the range of 5-10% (higher at younger ages). Can the authors estimate what fraction of the inception sample will have changed from private to public sector employment over the seven year followup?

As the authors will be aware, there is an extensive empirical literature, primarily in northern European settings, describing the relationship between sickness absence and subsequent risk of labour force exit via disability pension. As I reviewed the information in supplementary table 1, there is clear evidence that SA episodes in 2012 are more common among study subjects who were classified

	longitudinally to the DP cluster. That the relationship between sickness absence and subsequent risk of disability pension was not assessed is, in my assessment, a weakness. Overall, I am not confident this manuscript is novel or informative. As the authors note in the discussion, 'similar results were found in a previous cross-sectional study using the same data', a study by the same authors (Farrants K, Alexanderson K. Sickness absence among privately employed white-collar workers: A total population study in Sweden. Scand J Public Health. 2021 Mar;49(2):159-167. doi: 10.1177/1403494820934275.). Additionally, the authors note in the discussion that 'the results cannot directly be generalized to other types of occupational population or to other countries'. I find it regrettable that the study design did not enable a comparison between white collar workers in the public sectors and the private sectors.
--	--

VERSION 1 – AUTHOR RESPONSE

Reviewer: 1

Dr. Nada Maric, University of Banja Luka

Comment: First I would like to congratulate you on your excellent work. The Introduction is clearly written and leads to the core of the problem. The method is well described. The results are presented clearly. Discussion is good, you compare your results with other studies but I have only one suggestion. In discussion, you could more discuss about what could be a reason for these differences in sociodemographic characteristics between clusters and you could add implications for policymakers.

Best regards,

Author's response: Thank you very much for this review! Thank you also for your important suggestion. However, due to the descriptive nature of this study, it is very difficult to draw any implications for policymakers solely from the results of this study, without being speculative and unscientific. We hope that these results will inspire to further studies that together with this might provide scientific evidence in future systematic reviews.

Reviewer: 2

Prof. Cameron Mustard, Institute for Work and Health

Comment: This manuscript reports on an analysis of approximately 1,200,000 male and female white collar workers employed in the Swedish private sector. Data for the analysis was obtained from three administrative data sources. The study design is longitudinal, classifying employment status at baseline in 2012, and following workers over seven years to ascertain the incidence and duration of episodes of sickness absence (greater than 14 days) and the incidence of disability pension. A method termed 'sequence analysis' was used to derive five clusters of workers, and multinomial regression analysis was used to identify sociodemographic characteristics associated with cluster membership. Generally, the manuscript was well structured and frequently clearly written.

Author's response: Thank you very much for these comments!

Comment: The study rationale is expressed in the introduction, proposing that previous research on the incidence of sickness absence and disability pensions among white collar workers was limited. Additionally, the authors argue that studies of white collar workers in private sector employment are

also not extensive. While I am not persuaded that the authors' assessment that studies of white collar workers was limited is correct, I accept their assessment that there are limited longitudinal cohort studies of white collar workers in private sector employment. That said, I do think the introduction needs to make a stronger case for why a focus on private sector employment is justified.

Author's response:

Although we have decades of experience of research within the area, have searched the literature, asked numerous researchers within public health, insurance medicine, and occupational health, we have not been able to identify more studies on SA and/or DP among privately employed white collar workers than those referred to in the manuscript. If you can inform us about others we would be more than happy to include information about them in the introduction section of the manuscript.

Moreover, we agree that the Introduction about this could have been clearer, and therefore have now revised it line with your suggestions; see p 4 para 2-4.

Comment: The description of the sequence analysis methods is relatively brief. I would note that the potentially relevant information presented in Figure 1 (density plots over the seven year followup) was not interpreted for the reader. If this Figure is to be retained, please provide interpretation.

Author's response: This figure was included in order to give a description of the high variety of sequences and to give an indication of how the distribution of states varied over time in each of the clusters. We now provide an interpretation of this figure on p 8 para 2.

Comment: I was also puzzled that the authors did not consider evaluating cluster membership based on the cumulative burden of sickness absence. Specifically, in Cluster 2 and Cluster 3 it appeared to me that there would be value in distinguishing between low incidence/low days and high incidence / high days of sickness absence.

Author's response: In sequence analysis, one of the key issues is to first define the number of states as well as the content of those states. Having a large number of states makes the analyses unfeasible. Therefore, we decided on annual states (instead of shorter periods) of SA/DP net days and to distinguish SA by SA diagnoses in this exploratory first study of such sequences. We used having had any SA in the respective diagnosis categories or any DP during the studied year as the key quality to assign people to states for the year. It would, of course, be extremely interesting to distinguish between those with many and those with few days of SA in a year, but due to limitations in the method, that would most likely mean having to drop information about SA diagnosis from the analysis. The results from this study, particularly those you highlight from Cluster 2 and Cluster 3, could perhaps in the future inspire a different study, looking more in-depth into sequences of numbers of SA days in a year!

Comment: I was curious about the reliability of the SA measure adopted in this study, specifically physician-certified sickness absence episodes of more than 14 days that were paid by the Social Insurance Agency. Can the authors provide information on the proportion of SA episodes potentially eligible for coverage that are not reported to the Social Insurance Agency (and therefore not included in the MiDAS database)?

Author's response: The aim of this study was to explore future SA, regarding SA spells >14 days, and future DP in this occupational group. This measure is commonly used in studies conducted in the Nordic countries.

Regarding reliability of the used information on SA: as we used administrative data on state

administrated benefits (not self-reported, nor company reported) the reliability can be assumed to be very high. Information about all SA (and DP) benefits that the National Social Insurance Agency has approved of and paid to claimants must be reported to the government. Thus, the reliability regarding information on both SA spell start- and end-date as well as grade is considered to be very high. As we only have information on the shorter SA spells for those in the cohort who during the follow-up became unemployed (for a shorter or longer period), we, as stated in the manuscript, decided not to include information on these shorter SA spells in order not to introduce bias. As stated in the discussion section, there are pros and cons of not having access to information on the shorter SA spells. Of course, most SA spells are very short (one or a few days, e.g., SA due to a stomach flu, a cold, sprained ankle, etc.)—however, these spells do not contribute with many net days. Unfortunately, we did not have information about the shorter SA spells for most of the cohort.

However, we obtained very good information about all SA spells that were at least 15 days long (even if they were for part-time). Furthermore, we were able to include the net SA days during the first 14 days of those SA spells. This means that we had information on the longer SA spells that were validated by a physician and had a diagnosis.

Hopefully, other studies with other aims than ours might be able to collect information on shorter SA spells, and thus see if results differ.

Comment: I would also recommend additional details be included to clarify the source(s) of information underlying the status category 'ineligible for SA and DP'. My reading of the manuscript's methods suggests that this status category was derived by the authors from information in the LISA database, rather than an administrative determination in the MiDAS database.

Author's response: Thank you for alerting us on not being clear about this. You are right in that we did not have such information from MiDAS.

We discussed what to call this category, that is, the category of people in the cohort who were no longer eligible for SA or DP benefits, but have not come up with a better term.

The two main reasons for not being eligible for such benefits is having died or having emigrated from Sweden. We find it very important to include information about those not being eligible for the studied outcomes—otherwise it seems as if the group 'no SA/DP' is larger than is actually the case. Another reason for not being eligible for such benefits is not having high enough income from paid work or from parental-leave allowances, or unemployment benefits (about less than 1000 EUR/year) to be covered by the public SA benefit insurance—one reason for that could be e.g., old-age retirement. As suggested by the reviewer, we have such information from either LISA (regarding income and emigration) or the Cause of Death register (regarding year of death).

We have now further developed the information about this in the manuscript on p 6 para 1.

Comment: I found the definition of a white collar occupation to be insufficient. Readers are provided the following information: 'an occupational code according to the Swedish Standard for Occupational Classification indicating a white collar occupation'. No citation was provided. Additionally, the manuscript lacked a conceptual definition of a white collar occupation and did not, in my view, adequately describe how employment/working conditions may differ between men and women in this broad occupational category.

Author's response: We have now included a reference to the definition of white-collar occupations and how they were categorized. The definition of white-collar occupations covers a wide range of occupations in many different businesses as well as a wide range of working conditions. Our aim here was not to explore how working conditions differ between either white- and blue-collar workers, nor between different groups of white-collar workers, but rather to examine SA and DP over time among

privately employed white-collar workers.

As most studies, regarding most occupations, show large differences between women and men regarding SA and DP we also chose to stratify some of the analyses by sex. We have now included text about this in the Introduction section, to justify that part of our aim. The aim in this study was to explore if there were sex differences in SA and DP also in this group. Given that we did find such sex differences, future studies are needed to investigate possible reasons for that. Some reasons might be the ones you suggest, i.e., employment/working conditions, however, there are many other possibilities.

Comment: On the basis of the information provided in the manuscript's method section, I understand that private sector employment status was defined in 2012 as a baseline characteristic. If I am correct in this assumption, how is change from private to public sector employment addressed. In most high income countries, the annual incidence of job change (either through job leaving or job losing) is in the range of 5-10% (higher at younger ages). Can the authors estimate what fraction of the inception sample will have changed from private to public sector employment over the seven year followup?

Author's response: Yes, in all longitudinal cohort studies, people change situation—some die, some emigrate, some stop engaging in paid work, become self-employed, become old-age pensioned, and some change type of employer, type of job, etcetera. In this study, our aim was not to study change of job or change of type of employer, our aim was to follow the cohort prospectively regarding the defined specific outcomes—irrespective of future job or employer. In another part of the larger project that this study is a part of, published in a report only available in Swedish, we have showed that the majority were still privately employed white-collar workers in 2018, and of those that were not, the most common reason was that they had stopped working (1).

Comment: As the authors will be aware, there is an extensive empirical literature, primarily in northern European settings, describing the relationship between sickness absence and subsequent risk of labour force exit via disability pension. As I reviewed the information in supplementary table 1, there is clear evidence that SA episodes in 2012 are more common among study subjects who were classified longitudinally to the DP cluster. That the relationship between sickness absence and subsequent risk of disability pension was not assessed is, in my assessment, a weakness.

Author's response: You are absolutely correct that previous SA is a strong predictor for DP—as well as for subsequent SA. This is the reason why we also included DP as a state, and also why we included SA at baseline in the analyses in two ways. We assessed the association between SA at baseline and subsequent cluster membership both in terms of number of SA days during 2012, and as a series of binary variables indicating SA in various diagnoses in 2012. As you said, our study showed that there was a strong association between SA at baseline and subsequent cluster membership—both membership in the two clusters associated with subsequent SA, and the cluster characterized by DP, as shown both by high odds ratios for both ways of measuring SA in Table 2 and an uneven distribution in Supplementary Table 1.

However, it is also obvious from our results that most people on long-term SA did not end up with DP. Among employees, SA can be said to be a prerequisite for later being granted DP—that is, we did not expect anyone to have been granted DP, (full- or part-time) without a previous long-term SA.

Comment: Overall, I am not confident this manuscript is novel or informative. As the authors note in the discussion, 'similar results were found in a previous cross-sectional study using the same data', a study by the same authors (Farrants K, Alexanderson K. Sickness absence among privately employed white-collar workers: A total population study in Sweden. *Scand J Public Health*. 2021.49(2):159-167. doi: 10.1177/1403494820934275.).

Author's response: We would, in line with Reviewer 1, argue that there are several novelties to this

study. Some examples are the longitudinal design, that there hardly are any studies about future SA and DP in this large group at the labour market, that all workers in a whole country fulfilling the inclusion criteria could be included, without any drop outs, the very large cohort (1.3 million people), the use of high quality register data for both exposures and outcomes, the use of a rarely used method, i.e., sequence analysis, which meant that we were able to show the development of SA and DP through sequences of several different states over several years, rather than as a binary variable at a single time point at the end of follow-up, as is still most commonly done in prospective studies of SA/DP.

So far, there is no other study like this published.

Comment: Additionally, the authors note in the discussion that 'the results cannot directly be generalized to other types of occupational population or to other countries'.

Author's response: This text, regarding generalizability of the results to other countries or occupations, would be valid for any study about future SA and DP of a particular group. This is also one of the reasons for why such studies are conducted in different occupations, different countries and at different time periods. Based on results from them we can learn about differences and similarities and use that to draw conclusions about what are general phenomena and what are localized to particular contexts or time periods.

Comment: I find it regrettable that the study design did not enable a comparison between white collar workers in the public sectors and the private sectors.

Author's response: Thank you for this suggestion for another study with another aim and cohort. Hopefully this study will inspire such research.

1. Farrants K, Alexanderson K. Sjukfrånvaro bland privatanställda tjänstemän 2012-2018. Stockholm: Karolinska Institutet; 2022.